# Relating Leverage Scores and Density using Regularized Christoffel Functions

**Edouard Pauwels**
IRIT-AOC
Université Toulouse 3
Paul Sabatier
Toulouse, France

**Francis Bach**
INRIA
Ecole Normale Supérieure
PSL Research University
Paris, France

**Jean-Philippe Vert**
Google Brain
CBIO Mines ParisTech
PSL Research University
Paris, France

## Abstract

Statistical leverage scores emerged as a fundamental tool for matrix sketching and column sampling with applications to low rank approximation, regression, random feature learning and quadrature. Yet, the very nature of this quantity is barely understood. Borrowing ideas from the orthogonal polynomial literature, we introduce the *regularized Christoffel function* associated to a positive definite kernel. This uncovers a variational formulation for leverage scores for kernel methods and allows to elucidate their relationships with the chosen kernel as well as population density. Our main result quantitatively describes a decreasing relation between leverage score and population density for a broad class of kernels on Euclidean spaces. Numerical simulations support our findings.

## 1   Introduction

Statistical leverage scores have been historically used as a diagnosis tool for linear regression [16, 34, 10]. To be concrete, for a ridge regression problem with design matrix $X$ and regularization parameter $\lambda > 0$, the leverage score of each data point is given by the diagonal elements of $X(X^\top X + \lambda I)^{-1}X^\top$. These leverage scores characterize the importance of the corresponding observations and are key to efficient subsampling with optimal approximation guarantees. Therefore, leverage scores emerged as a fundamental tool for matrix sketching and column sampling [22, 21, 13, 36], and play an important role in low rank matrix approximation [11, 6], regression [2, 28, 20], random feature learning [29] and quadrature [7]. The notion of leverage score is seen as an intrinsic, setting-dependent quantity, and should eventually be estimated. In this work we elucidate a relation between leverage score and the learning setting (population measure and statistical model) when used with kernel methods.

For that purpose, we introduce a variant of the *Christoffel function*, a classical tool in polynomial algebra which provides a bound for the evaluation at a given point of a given degree polynomial $P$ in terms of an average value of $P^2$. The Christoffel function is an important object in the theory of orthogonal polynomials [32, 14] and found applications in approximation theory [26] and spectral analysis of random matrices [5]. It is parametrized by the degree of polynomials considered and an associated measure, and we know that, as the polynomial degree increases, it encodes information about the support and the density of its associated measures, see [23, 24, 33] for the univariate case and [8, 9, 37, 38, 18, 19] in the multivariate case.

The variant we propose amounts to replacing the set of polynomials with fixed degree, used in the definition of the Christoffel function, by a set of function with bounded norm in a reproducing kernel Hilbert space (RKHS)[1]. More precisely, given a density $p$ on $\mathbb{R}^d$ and a regularization parameter

$\lambda > 0$, we introduce $C_\lambda : \mathbb{R}^d \to \mathbb{R}$, the *regularized Christoffel function* where $\lambda$ plays a similar role as the degree for polynomials. The function $C_\lambda$ turns out to have intrinsic connections with statistical leverage scores, as the quantity $1/C_\lambda$ corresponds precisely to a notion of leverage used in [6, 2, 28, 7]. As a consequence, we uncover a variational formulation for leverage scores which helps elucidate connections with the RKHS and the density $p$ on $\mathbb{R}^d$.

Our main contribution is a precise asymptotic expansion of $C_\lambda$ as $\lambda \to 0$ under restrictions on the RKHS. To give a concrete example, if we consider the Sobolev space of functions on $\mathbb{R}^d$ with squared integrable derivatives of order up to $s > d/2$, we obtain, the asymptotic equivalent

$$C_\lambda(z) \underset{\lambda \to 0, \, \lambda > 0}{\sim} q_0^{-1} \lambda^{d/(2s)} p(z)^{1-d/2s},$$

for $z$ a continuity point of $p$ with $p(z) > 0$. Here $q_0$ is an explicit constant which only depends on the RKHS. We recover scalings with respect to $\lambda$ which matches known estimates for the usual degrees of freedom [28, 7]. More importantly, we also obtain a precise spatial description of $C_\lambda(z)$ (i.e., dependency on $z$), and deduce that the leverage score is itself proportional to $p(z)^{d/(2s)-1}$ in the limit. Roughly speaking, large scores are given to low density regions (note that $d/(2s) - 1 < 0$). This result has several potential consequences for machine learning:

(i) The Christoffel function could be used for density or support estimation. This has connections with the spectral approach proposed in [35] for support learning. (ii) This could provide a more efficient way to estimate leverage scores through density estimation. (iii) When leverage scores are used for sampling, the required sample size depends on the ratio between the maximum and the average leverage scores [28, 7]. Our results imply that this ratio can be large if there exists low-density regions, while it remains bounded otherwise.

**Organization of the paper.** We introduce the regularized Christoffel function in Section 2 and explicit connections with leverage scores and orthogonal polynomials. Our main result and assumptions are described in abstract form in Section 3, they are presented as a general recipe to compute asymptotic expansions for the regularized Christoffel function. Section 3.3 describes an explicit example and a precise asymptotic for an important class of RKHS related to Sobolev spaces. We illustrate our results numerically in Section 4. The proofs are postponed to Appendix B while Appendix A contains additional properties and simulations, and Appendix C contains further lemmas.

**Notations.** Let $d$ denote the ambient dimension, $\mathbf{0}$ denote the origin in $\mathbb{R}^d$ and $C(\mathbb{R}^d), L^1(\mathbb{R}^d), L^2(\mathbb{R}^d), L^\infty(\mathbb{R}^d)$ denote the complex-valued function on $\mathbb{R}^d$ which are respectively continuous, absolutely integrable, square integrable, measurable and essentially bounded. For any $f \in L^1(\mathbb{R}^d)$, let $\hat{f} : \mathbb{R}^d \mapsto \mathbb{C}$ be its Fourier transform, $\hat{f} : \omega \mapsto \int_{\mathbb{R}^d} f(x)e^{-ix^\top \omega}dx$. For $g \in L^1(\mathbb{R}^d)$, its inverse Fourier transform is $x \mapsto \frac{1}{(2\pi)^d} \int_{\mathbb{R}^d} g(x)e^{ix^\top \omega}d\omega$. If $f \in L^1(\mathbb{R}^d) \cap C(\mathbb{R}^d)$ and $\hat{f} \in L^1(\mathbb{R}^d)$, then inverse transform composed with direct transform leaves $f$ unchanged. The Fourier transform is extended to $L^2(\mathbb{R}^d)$ by a density argument. It defines an isometry: if $f \in L^2(\mathbb{R}^d)$, Parseval formula writes $\int_{\mathbb{R}^d} |f(x)|^2 dx = \frac{1}{(2\pi)^d} \int_{\mathbb{R}^d} |\hat{f}(\omega)|^2 d\omega$. See, *e.g.*, [17, Chapter 11].

We identify $x$ with a set of $d$ real variables $x_1, \ldots, x_d$. We associate to a multi-index $\beta = (\beta_i)_{i=1,\ldots,d} \in \mathbb{N}^d$ the monomial $x^\beta := x_1^{\beta_1} x_2^{\beta_2} \ldots x_d^{\beta_d}$ whose degree is $|\beta| := \sum_{i=1}^d \beta_i$. The linear span of monomials forms the set of $d$-variate polynomials. The degree of a polynomial is the highest of the degrees of its monomials with nonzero coefficients (null for the null polynomial). A polynomial $P$ is said to be homogeneous of degree $2s \in \mathbb{N}$ if for all $\lambda \in \mathbb{R}$, $x \in \mathbb{R}^d$, $P(\lambda x) = \lambda^{2s} P(x)$, it is then composed only of monomials of degree $2s$. See [14] for further details.

## 2 Regularized Christoffel function

### 2.1 Definition

In what follows, $k$ is a positive definite, continuous, bounded, integrable, real-valued kernel on $\mathbb{R}^d \times \mathbb{R}^d$ and $p$ is an integrable real function over $\mathbb{R}^d$. We denote by $\mathcal{H}$ the RKHS associated to $k$ which is assumed to be dense in $L^2(p)$, the normed space of functions, $f : \mathbb{R}^d \mapsto \mathbb{R}$, such that $\int_{\mathbb{R}^d} f^2(x)p(x)dx < +\infty$. This will be made more precise in Section 3.1.

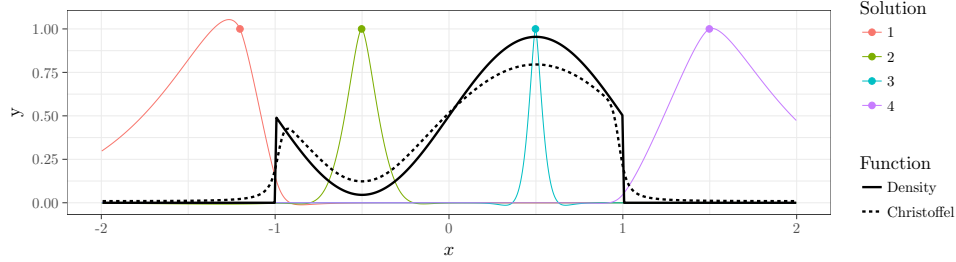

Figure 1: The black lines represent a density and the corresponding Christoffel function. The colored lines are solutions of problem in (1), the corresponding $z$ being represented by the dots. Outside the support, the optimum is smooth and high values have small overlap with the support. Inside the support, the optimum is less smooth, it forms a peak, sharper in higher density regions.

**Definition 1** *The* regularized Christoffel function, *is given for any* $\lambda > 0$, $z \in \mathbb{R}^d$ *by*

$$C_{\lambda,p,k}(z) = \inf_{f \in \mathcal{H}} \int_{\mathbb{R}^d} f(x)^2 p(x) dx + \lambda \|f\|_{\mathcal{H}}^2 \quad \text{such that} \quad f(z) = 1 \,. \tag{1}$$

If there is no confusion about the kernel $k$ and the density $p$, we will use the notation $C_\lambda = C_{\lambda,p,k}$. More compactly, setting for $z \in \mathbb{R}^d$, $\mathcal{H}_z = \{f \in \mathcal{H}, f(z) = 1\}$, we have $C_\lambda \colon z \mapsto \inf_{f \in \mathcal{H}_z} \|f\|_{L^2(p)}^2 + \lambda \|f\|_{\mathcal{H}}^2$. The value of (1) is intuitively connected to the density $p$. Indeed, the constraint $f(z) = 1$ forces $f$ to remains far from zero in a neighborhood of $z$. Increasing the $p$ measure of this neighborhood increases the value of the Christoffel function. In low density regions, the constraint has little effect which allows to consider smoother functions with little overlap with higher density regions and decreases the overall cost. An illustration is given in Figure 1.

## 2.2 Relation with orthogonal polynomials

The name *Christoffel* is borrowed from the orthogonal polynomial literature [32, 14, 26]. In this context, the Christoffel function is defined as follows for any degree $l \in \mathbb{N}$:

$$\Lambda_l \colon z \mapsto \min_{P \in \mathbb{R}_l[X]} \int (P(x))^2 p(x) dx \quad \text{such that} \quad P(z) = 1 \,,$$

where $\mathbb{R}_l[X]$ denotes the set of $d$ variate polynomials of degree at most $l$. The regularized Christoffel function in (1) is a direct extension, replacing the polynomials of increasing degree by functions in a RKHS with increasing norm. $\Lambda_l$ has connections with quadrature and interpolation [26], potential theory and random matrices [5], orthogonal polynomials [26, 14]. Relating the asymptotic for large $l$ and properties of $p$ has also been a long lasting subject of research [23, 24, 33, 8, 9, 37, 38, 18, 19]. The idea of studying the relation between $C_\lambda$ and $p$ was directly inspired by these works.

## 2.3 Relation with leverage scores for kernel methods

The (non-centered) covariance of $p$ on $\mathcal{H}$ is the bilinear form Cov $\colon \mathcal{H} \times \mathcal{H} \to \mathbb{R}$ given by:

$$\forall (f,g) \in \mathcal{H}^2 \,, \quad \text{Cov}(f,g) = \int_{\mathbb{R}^d} f(x)g(x)p(x)dx \,.$$

The covariance operator $\Sigma \colon \mathcal{H} \to \mathcal{H}$ is then defined such that for all $f, g \in \mathcal{H}$, $\text{Cov}(f, g) = \langle \Sigma f, g \rangle_{\mathcal{H}}$. If $\Sigma$ is bounded with respect to $\|\cdot\|_{\mathcal{H}}$, then Lemma 5 in Appendix C shows that:

$$\forall z \in \mathbb{R}^d, \quad C_\lambda(z) = \left( \langle k(z, \cdot), (\Sigma + \lambda I)^{-1} k(z, \cdot) \rangle_{\mathcal{H}} \right)^{-1} \,,$$

which provides a direct link with leverage scores [7], as $C_\lambda(z)$ is exactly the inverse of the population leverage score at $z$.

As $\lambda \to 0$, we typically have $\langle k(z, \cdot), (\Sigma + \lambda I)^{-1} k(z, \cdot) \rangle_{\mathcal{H}} \to +\infty$. It is worth emphasizing that spectral estimators (with other functions of the covariance operator than $(\Sigma + \lambda I)^{-1}$) have been proposed for support inference in [35]. An example of such estimator has the form $F_\lambda \colon z \mapsto \langle k(z, \cdot), \Sigma(\Sigma + \lambda I)^{-1} k(z, \cdot) \rangle_{\mathcal{H}}$, for which finite level sets encode information about the support of $p$ as $\lambda \to 0$ [35]. Our main result should extend to broader classes of spectral functions.

## 2.4 Estimation from a discrete measure

Practical computation of the regularized Christoffel function requires to have access to the covariance operator $\Sigma$, which is not available in closed form in general. A plugin solution consists in replacing integration with weight $p$ by a discrete approximation of the form $d\rho_n = \sum_{i=1}^n \eta_i \delta_{x_i}$, where for each $i = 1, \ldots, n$, $\eta_i \in \mathbb{R}_+$ is a weight, $x_i \in \mathbb{R}^d$ and $\delta_{x_i}$ denotes the Dirac measure at $x_i$. We may assume without loss of generality that the points are distinct. Given a kernel function $k$ on $\mathbb{R}^d \times \mathbb{R}^d$, let $K = (k(x_i, x_j))_{i,j=1,\ldots,n} \in \mathbb{R}^{n \times n}$ be the Gram matrix and $K_i$ the $i$-th column of $K$ for $i = 1, \ldots, n$. We have a closed form expression for the Christoffel function with plug-in measure $d\rho_n$, for any $\lambda > 0$, $i = 1, \ldots, n$:

$$C_{\lambda,\rho_n,k}(x_i) = \inf_{f(x_i)=1} \frac{1}{n} \sum_{j=1}^n \eta_j (f(x_j))^2 + \lambda \|f\|_{\mathcal{H}}^2 = \left( K_i^\top \left( K \operatorname{diag}(\eta) K + \lambda K \right)^{-1} K_i \right)^{-1}. \quad (2)$$

This is a consequence of the representer theorem [30]; Lemma 5 allows to deal with the constraint explicitly. Note that if $\eta_i > 0$ for all $i = 1, \ldots, n$, then the Christoffel function may be obtained as a weighted diagonal element of a smoothing matrix, as for all $i$, thanks to the matrix inversion lemma, $K_i^\top \left( K \operatorname{diag}(\eta) K + \lambda K \right)^{-1} K_i = \eta_i^{-1} \left( K (K + \lambda \operatorname{Diag}(\eta)^{-1})^{-1} \right)_{ii}$. This draws an important connection with statistical leverage score [21, 13] as it corresponds to the notion introduced for kernel ridge regression [6, 2, 28]. It remains to choose $\eta$ so that $d\rho_n$ approximates integration with weight $p$.

**Monte Carlo approximation:** Assuming that $\int_{\mathbb{R}^d} p(x) dx = 1$, if one has the possibility to draw an *i.i.d.* sample $(x_i)_{i=1,\ldots,n}$, with density $p$, then one can use $\eta_i = \frac{1}{n}$ for $i = 1, \ldots, n$. The quality of this approximation is of order $\lambda^{-2} n^{-1/2}$ (see Appendix A). If $\lambda^2 n^{1/2}$ is large enough, then we obtain a good estimation of the Christoffel function (note that better bounds could be obtained with respect to $\lambda$ using tools from [6, 2, 28]).

**Riemann sums:** If the density $p$ is piecewise smooth, one can approximate integrals with weight $p$ by using a uniform grid and a Riemann sum with weights proportional to $p$. The bound in Eq. (8) also holds, the quality of this approximation is typically of the order of $n^{-1/d}$ which is attractive in dimension 1 but quickly degrades in larger dimensions.

Depending on properties of the integrand, quasi Monte Carlo methods could yeld faster quadrature rules [12], more quantitative deviation bounds and faster rates is left for future research.

## 3 Relating regularized Christoffel functions to density

We first make precise our notations and assumptions in Section 3.1 and describe our main result in Section 3.2 using Assumption 2 which is given in abstract form. We then describe how this assumption is satisfied by a broad class of kernels in Section 3.3.

### 3.1 Assumptions

**Assumption 1**

1. *The kernel $k$ is translation invariant: for any $x, y \in \mathbb{R}^d$, $k(x, y) = q(x - y)$ where $q \in L^1(\mathbb{R}^d)$ is the inverse Fourier transform of $\hat{q} \in L^1(\mathbb{R}^d)$ which is real valued and strictly positive.*

2. *The density $p \in L^1(\mathbb{R}^d) \cap L^\infty(\mathbb{R}^d)$ is finite and nonnegative everywhere.*

Under Assumption 1, $k$ is a positive definite kernel by Bochner's theorem and we have an explicit characterization of the associated RKHS (see *e.g.* [35, Proposition 4]),

$$\mathcal{H} = \left\{ f \in C(\mathbb{R}^d) \cap L^2(\mathbb{R}^d); \int_{\mathbb{R}^d} \frac{|\hat{f}(\omega)|^2}{\hat{q}(\omega)} d\omega < +\infty \right\}, \quad (3)$$

with inner product

$$\langle \cdot, \cdot \rangle_{\mathcal{H}} : (f, g) \mapsto \frac{1}{(2\pi)^d} \int_{\mathbb{R}^d} \frac{\hat{f}(\omega)\bar{\hat{g}}(\omega)}{\hat{q}(\omega)} d\omega . \tag{4}$$

**Remark 1** *The assumption that $\hat{q} \in L^1(\mathbb{R}^d)$ implies by the Riemann-Lebesgue theorem that $q$ is in $C_0(\mathbb{R}^d)$, the set of continuous functions vanishing at infinity. Since $\hat{q}$ is strictly positive, its support is $\mathbb{R}^d$ and [31, Proposition 8] implies that $k$ is $c_0$-universal, i.e., that $\mathcal{H}$ is dense in $C_0(\mathbb{R}^d)$ w.r.t. the uniform norm. As a result, $\mathcal{H}$ is also dense in $L^2(d\rho)$ for any probability measure $\rho$.*

**Remark 2** *For any $f \in \mathcal{H}$, we have by Cauchy-Schwartz inequality*

$$\left( \int_{\mathbb{R}^d} |\hat{f}(\omega)| d\omega \right)^2 \leq \int_{\mathbb{R}^d} \frac{|\hat{f}(\omega)|^2}{\hat{q}(\omega)} d\omega \int_{\mathbb{R}^d} \hat{q}(\omega) d\omega,$$

*and the last term is finite by Assumption 1. Hence $\hat{f} \in L^1(\mathbb{R}^d)$ and we have $f(0) = \int_{\mathbb{R}^d} \hat{f}$ where the integral is understood in the usual sense. In this setting any $f \in \mathcal{H}$ is uniquely determined everywhere on $\mathbb{R}^d$ by its Fourier transform and we have for any $f \in \mathcal{H}$, $\|f\|_{L^\infty(\mathbb{R}^d)} \leq \|\hat{f}\|_{L^1(\mathbb{R}^d)} \leq \|f\|_{\mathcal{H}} \sqrt{q(0)}$.*

## 3.2   Main result

Problem (1) is related to a simpler variational problem with explicit solution. For any $\lambda > 0$, let

$$D(\lambda) := \min_{f \in \mathcal{H}} \int_{\mathbb{R}^d} f(x)^2 dx + \lambda \|f\|_{\mathcal{H}}^2 \text{ subject to } f(\mathbf{0}) = 1. \tag{5}$$

Note that $D(\cdot)$ does not depend on $p$ and corresponds to the Christoffel function at the origin $\mathbf{0}$, or any other points by translation invariance, for the Lebesgue measure on $\mathbb{R}^d$. The solutions of (5) have an explicit description which proof is presented in Appendix B.2.

**Lemma 1** *For any $\lambda > 0$, $D(\lambda) = \frac{(2\pi)^d}{\int_{\mathbb{R}^d} \frac{\hat{q}(\omega)}{\lambda + \hat{q}(\omega)} d\omega}$, and this value is attained by the function*

$$f_\lambda : x \mapsto D(\lambda) \frac{1}{(2\pi)^d} \int_{\mathbb{R}^d} \frac{\hat{q}(\omega) e^{i\omega^\top x}}{\hat{q}(\omega) + \lambda} d\omega.$$

**Remark 3** *We directly obtain $D(\lambda) \geq \frac{(2\pi)^d \lambda}{q(0)}$, for any $\lambda > 0$. Finally, let us mention that Assumption 1 ensures that $\lim_{\lambda \to 0} D(\lambda) = 0$ as $\int_{\mathbb{R}^d} \frac{\hat{q}(\omega)}{\lambda + \hat{q}(\omega)} d\omega \geq \int_{\hat{q}(\omega) \geq \lambda} \frac{d\omega}{2}$ which diverges as $\lambda \to 0$.*

We denote by $g_\lambda$ the inverse Fourier transform of $\frac{\hat{q}}{\lambda + \hat{q}}$, i.e., $g_\lambda = f_\lambda / D(\lambda)$. It satisfies $g_\lambda(0) = \frac{1}{D(\lambda)}$. Intuitively, as $\lambda$ tends to 0, $g_\lambda$, should be approaching a Dirac in the sense that $g_\lambda$ tends to 0 everywhere except at the origin where it goes to $+\infty$. The purpose of the next Assumption is to quantify this intuition.

**Assumption 2** *For the kernel $k$ given in Assumption 1 and $f_\lambda$ given in Lemma 1, there exists $\varepsilon \colon \mathbb{R}_+ \to \mathbb{R}_+$ such that, as $\lambda \to 0$, $\varepsilon(\lambda) \to 0$, and*

$$\int_{\|x\| \geq \varepsilon(\lambda)} f_\lambda^2(x) dx = o(\lambda D(\lambda)).$$

See Section 3.3 for specific examples. We are now ready to describe the asymptotic inside the support of $p$, the proof is given in Appendix B.1.

**Theorem 1** *Let $q$, $k$ and $p$ be given as in Assumption 1 and let $C_\lambda$ be defined as in (1). If Assumption 2 holds, then, for any $z \in \mathbb{R}^d$ such that $p(z) > 0$ and $p$ is continuous at $z$, we have*

$$C_\lambda(z) \underset{\lambda \to 0, \, \lambda > 0}{\sim} p(z) D\left( \frac{\lambda}{p(z)} \right).$$

**Proof sketch.** The equivalent is shown by using the variational formulation in Eq. (1). A natural candidate for the optimal function $f$ is the optimizer obtained from Lebesgue measure in Eq. (5), scaled by $p(z)$. Together with Assumption 2, this leads to the desired upper bound. In order to obtain the corresponding lower bound, we consider Lebesgue measure restricted to a small ball around $z$. Using linear algebra and expansions of operator inverses, we relate the optimal value directly to the optimal value $D(\lambda)$ of Eq. (5).

This result is complemented by the following which describes the asymptotic behavior outside the support of $p$, the proof is given in Appendix B.3.

**Theorem 2** *Let $q$, $k$ and $p$ be given as in Theorem 1. Then, for any $z \in \mathbb{R}^d$, such that there exists $\epsilon > 0$ with $\int_{\|z-x\| \leq \epsilon} p(x) dx = 0$, we have*

$$(i) \qquad C_\lambda(z) \underset{\lambda \to 0,\, \lambda > 0}{=} O(\sqrt{\lambda} D(\sqrt{\lambda})).$$

*If furthermore there exists $a \geq 0$ and $c > 0$ such that, for any $\omega \in \mathbb{R}^d$, $\hat{q}(\omega) \geq \frac{c}{1+\|\omega\|^a}$ , then, for any such $z \in \mathbb{R}^d$, we have*

$$(ii) \qquad C_\lambda(z) \underset{\lambda \to 0,\, \lambda > 0}{=} O(\lambda).$$

**Proof sketch.** Since only an upper-bound is needed, we simply have to propose a candidate function for $f$, and we build one from the solution of Eq. (5) for (i) and directly from properties of kernels for (ii).

**Remark 4** *Theorems 1 and 2 underline separation between the "inside" and the "outside" of the support of $p$ and describes the fact that the convergence to $0$ as $\lambda$ decreases is faster outside: (i), if $\log(D(\lambda)) = \alpha \log(\lambda) + o(1)$ with $\alpha < 1$ (which is the case in most interesting situations), then $C_\lambda(z) = O(\sqrt{\lambda} D(\sqrt{\lambda})) = o(D(\lambda))$. (ii), it holds that $\lambda = o(D(\lambda))$. Hence in most cases, the values of the Christoffel function outside of the support of $p$ are negligible compared to the ones inside the support of $p$.*

Combining Theorem 1 and 2 does not describe what happens in the limit case where neither of the conditions on $z$ hold, for example on the boundary of the support or at discontinuity points of the density. We expect that this highly depends on the geometry of $p$ and its support. In the polynomial case on the simplex, the rate depends on the dimension of the largest face containing the point of interest [38]. Settling down this question in the RKHS setting is left for future research.

### 3.3 A general construction

We describe a class of kernels for which Assumptions 1 and 2 hold, and Theorem 1 can be applied, which includes Sobolev spaces. We also compute explicit equivalents for $D(\cdot)$ in (5). We first introduce a definition and an assumption.

**Definition 2** *For any $s \in \mathbb{N}^*$, a $d$-variate polynomial $P$ of degree $2s$ is called $2s$-positive if it satisfies the following.*

- *Let $Q$ denote the $2s$-homogeneous part of $P$ (the sum of its monomial of degree $2s$). $Q$ is (strictly) positive on the unit sphere in $\mathbb{R}^d$.*

- *The polynomial $R = P - Q$ satisfies $R(x) \geq 1$ for all $x \in \mathbb{R}^d$.*

**Remark 5** *If $P$ is $2s$-positive, then it is always greater than $1$ and its $2s$-homogeneous part is strictly positive except at the origin. The positivity of $Q$ forbids the use of polynomial $P$ of the form $\prod_{i=1}^d (1 + w_i^2)$ which would allow to treat product kernels. Indeed, this would lead to $Q(\omega) = \prod_{i=1}^d w_i^2$ which is not positive on the unit sphere. The last condition on $R$ is not very restrictive as it can be ensured by a proper rescaling of $P$ if we have $R > 0$ only.*

**Assumption 3** *Let $P$ be a $2s$-positive, $d$-variate polynomial and let $\gamma \geq 1$ be such that $2s\gamma > d$. The kernel $k$ is given as in Assumption 1 with $\hat{q} = \frac{1}{P^\gamma}$.*

One can check that $q$ in Assumption 3 is well defined and satisfies Assumption 1. A famous example of such a kernel is the Laplace kernel $(x, y) \mapsto e^{-\|x-y\|}$ which amounts, up to a rescaling, to choose $P$ of the form $1 + a\| \cdot \|^2$ for $a > 0$ and $\gamma = \frac{d+1}{2}$. In addition, Assumption 3 allows to capture the usual multi-dimensional Sobolev space of functions with square integrable partial derivatives up to order $s$, with $s > 2/d$, and the corresponding norm. We now provide the main result of this section.

**Lemma 2** *Assume that $p$ and $k$ are given as in Assumption 1 and 3. Then Assumption 2 is satisfied. More precisely, set $q_0 = \frac{1}{(2\pi)^d} \int_{\mathbb{R}^d} \frac{1}{1+Q(\omega)^\gamma} d\omega$ and $p = \lceil s\gamma \rceil$, then for any $l < \left(1 - \frac{d}{2s\gamma}\right)/(8p)$ the following holds true as $\lambda \to 0$, $\lambda > 0$ :*

$$(i) \quad D(\lambda) \sim \frac{\lambda^{\frac{d}{2s\gamma}}}{q_0}, \qquad\qquad (ii) \quad \int_{\|x\| \geq \lambda^l} f_\lambda^2(x) dx = o\left(\lambda D(\lambda)\right).$$

**Remark 6** *If $Q \colon \omega \mapsto \|\omega\|^{2s}$, using spherical coordinate integration, we obtain*

$$q_0 = \frac{1}{(2\pi)^d} \int_{\mathbb{R}^d} \frac{1}{1+Q(\omega)^\gamma} d\omega = \frac{1}{2^{d-1}\pi^{\frac{d}{2}}\Gamma\left(\frac{d}{2}\right)} \int_0^{+\infty} \frac{r^{d-1}}{1+r^{2s\gamma}} d\omega = \frac{1}{2^{d-1}\pi^{\frac{d}{2}}\Gamma\left(\frac{d}{2}\right)} \frac{\pi}{2s\gamma \sin\left(\frac{d\pi}{2s\gamma}\right)}.$$

The proof is presented in Appendix B.4. We have the following corollary which is a direct application of Theorem 1. It explicits the asymptotic for the Christoffel function, in terms of the density $p$.

**Corollary 1** *Assume that $p$ and $k$ are given as in Assumption 1 and 3 and that $z \in \mathbb{R}^d$ is such that $p(z) > 0$ and $p$ is continuous at $z$. Then as $\lambda \to 0$, $\lambda > 0$,*

$$C_\lambda(z) \sim \lambda^{\frac{d}{2s\gamma}} p(z)^{1-\frac{d}{2s\gamma}} \frac{1}{q_0}.$$

# 4 Numerical illustration

In this section we provide numerical evidence confirming the rate described in Corollary 1. We use the Matérn kernel, a parametric radial kernel allowing different values of $\gamma$ in Assumption 3.

## 4.1 Matérn kernel

We follow the description of [27, Section 4.2.1], note that the Fourier transform is normalized differently in our paper. For any $\nu > 0$ and $l > 0$, we let for any $x \in \mathbb{R}^d$,

$$q_{\nu,l}(x) = \frac{2^{1-\nu}}{\Gamma(\nu)} \left(\frac{\sqrt{2\nu}\|x\|}{l}\right)^\nu K_\nu\left(\frac{\sqrt{2\nu}\|x\|}{l}\right), \qquad (6)$$

where $K_\nu$ is the modified Bessel function of the second kind [1, Section 9.6]. This choice of $q$ satisfies Assumption 3, with $s = 1$ and $\gamma = \nu + \frac{d}{2}$. Indeed, for any $\nu, l > 0$, its Fourier transform is given for any $\omega \in \mathbb{R}^d$

$$\hat{q}_{\nu,l}(\omega) = \frac{2^d \pi^{\frac{d}{2}} \Gamma\left(\nu + d/2\right)(2\nu)^\nu}{\Gamma(\nu)l^{2\nu}} \frac{1}{\left(\frac{2\nu}{l^2} + \|\omega\|^2\right)^{\nu+\frac{d}{2}}}. \qquad (7)$$

## 4.2 Empirical validation of the convergence rate estimate

Corollary 1 ensures that, given $\nu, l > 0$ and $q$ in (6), as $\lambda \to 0$, we have for appropriate $z$, $C_\lambda(z) \sim \lambda^{\frac{d}{2\nu+d}} p(z)^{\frac{2\nu}{2\nu+d}} / q_0(\nu, l)$. We use the Riemann sum plug-in approximation described in Section 2.4 to illustrate this result numerically. We perform extensive investigations with compactly supported sinusoidal density in dimension 1. Note that from Remark 6 we have the closed form expression

$$q_0(\nu, l) = \left(\frac{2^d \pi^{\frac{d}{2}} \Gamma(\nu+d/2)(2\nu)^\nu}{\Gamma(\nu)l^{2\nu}}\right)^{\frac{1}{2\nu+d}} \frac{1}{(2\nu+d) \sin\left(\frac{d\pi}{2\nu+d}\right)}.$$

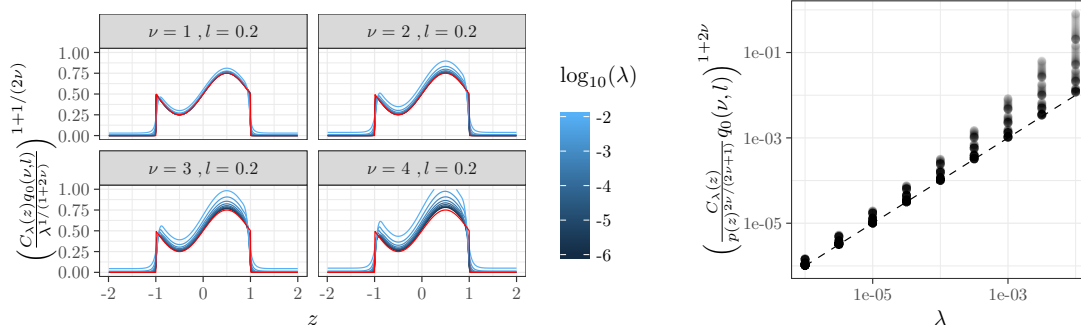

Figure 2: The target density $p$ represented in red. We consider different choices of $\nu$ and $l$ for $q$ as in (6). We use the Riemann sum plug-in approximation described in (2) with $n = 2000$. *Left:* the fact that the estimate is close to the density is clear for small values of $\lambda$. *Right:* the dotted line represents the identity. This suggests that the rate estimate is of the correct order in $\lambda$.

**Relation with the density:** For a given choice of $\nu, l > 0$, as $\lambda \to 0$, we should obtain for appropriate $z$ that the quantity, $\left( \frac{C_\lambda(z) q_0(\nu,l)}{\lambda^{d/(d+2\nu)}} \right)^{1+d/(2\nu)}$ is roughly equal to $p(z)$. This is confirmed numerically as presented in Figure 2 (left), for different choices of the parameters $\nu$.

**Convergence rate:** For a given choice of $\nu, l > 0$, as $\lambda \to 0$, we should obtain for appropriate $z$ that the quantity $\left( \frac{C_\lambda(z)}{p(z)^{2\nu/(2\nu+1)}} q_0(\nu,l) \right)^{\frac{2\nu+d}{d}}$ is roughly equal to $\lambda$. Considering the same experiment confirms this finding as presented in Figure 2 right, which suggests that the exponent in $\lambda$ is of the correct order.

**Additional experiments:** A piecewise constant density is considered in Appendix A which also contains simulations suggesting that the asymptotic has a different nature for the Gaussian kernel for which we conjecture that our result does not hold.

## 5 Conclusion and future work

We have introduced a notion of Christoffel function in RKHS settings. This allowed to derive precise asymptotic expansions for a quantity known as statistical leverage score which has a wide variety of applications in machine learning with kernel methods. Our main result states that the leverage score is inversely proportional to a power of the population density at the considered point. This has intuitive meaning as leverage score is a measure of the contribution of a given observation to a statistical estimate. For densely populated region, a specific observation, which should have many close neighbors, has less effect on a statistical estimate than observations in less populated areas of space. Our observation gives a precise meaning to this statement and sheds new light on the relevance of the notion of leverage score. Furthermore, it is coherent with known results in the orthogonal polynomial literature from which the notion of Christoffel function was inspired.

Direct extensions of this work include approximation bounds for our proposed plug-in estimate and tuning of the regularization parameter $\lambda$. A related question is the relevance of the proposed variational formulation for the statistical estimation of leverage scores when learning from random features, in particular random Fourier features and density/support estimation. Another line of future research would be the extension of our estimates to broader classes of RKHS, for example, kernels with product structure, such as the $\ell_1$ counterpart of the Laplace kernel. Finally, it would be interesting to extend the concepts to abstract topological spaces beyond $\mathbb{R}^d$.

**Acknowledgements**

We acknowledge support from the European Research Council (grant SEQUOIA 724063).

## Footnotes

[1]Kernelized Christoffel functions were first proposed by Laurent El Ghaoui and independently studied in [4].

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
