[Supplementary Material]

This is the supplementary material for the paper: "Relating Leverage Scores and Density using Regularized Christoffel Functions".

## A   Additional properties and numerical simulations

**Monotonicity properties.**   It is obvious from the definition in (1) that the regularized Christoffel function is an increasing function of $\lambda$, it is also concave. If $p$ and $\tilde{p}$ are as in Assumption 1.2, for any $\lambda > 0$, $z \in \mathbb{R}^d$

$$C_{\lambda,p+\tilde{p},k}(z) = \inf_{f \in \mathcal{H}_z} \|f\|^2_{L^2(p+\tilde{p})} + \lambda\|f\|^2_{\mathcal{H}} \geq \inf_{f \in \mathcal{H}_z} \|f\|^2_{L^2(p)} + \lambda\|f\|^2_{\mathcal{H}} = C_{\lambda,p,k}(z),$$

that is, the regularized Christoffel function is an increasing function of the underlying density.

The Christoffel function is also monotonic with respect to kernel choice. For any two positive definite kernels $k$ and $k'$, we have for any $\lambda > 0$,

$$C_{\lambda,p,k+k'} \leq C_{\lambda,p,k'},$$

that is, the regularized Christoffel function is a decreasing function of the underlying kernel. Indeed, for any positive definite kernels $k$ and $k'$, denote by $\mathcal{H}$ the RKHS associated to $k$ and $\tilde{\mathcal{H}}$ the RKHS associated to $k + k'$. We have $\mathcal{H} \subset \tilde{\mathcal{H}}$ and $\|\cdot\|_{\tilde{\mathcal{H}}} \leq \|\cdot\|_{\mathcal{H}}$ [3, Section 7, Theorem I].

**Overfitting.**   We are interested in the asymptotic behavior of the Christoffel function as the regularization parameter $\lambda$ tends to 0. This is approximated based on $n$ points using the plug-in approach in Section 2.4. For a fixed value of $n$, the empirical measure $d\rho_n$ is supported on only $n$ points and the asymptotic as $\lambda \to 0$ is straightforward. For example if Theorem 2 *(ii)* holds, then we obtain $O(\lambda)$ outside of the support and $\eta_i$ at each support point $x_i$, $i = 1, \ldots, n$. This is because the quality of approximation of $p$ by $d\rho_n$ depends on the regularity of the corresponding test functions. Small values of the regularization parameter $\lambda$ allow to consider functions with very low regularity so that the approximation become vacuous and the obtained estimate only reflects the finiteness of the support of $d\rho_n$. This phenomenon is illustrated in Figure 3. Hence, when using the proposed plug-in approach, it is fundamental to carefully tune the considered value of $\lambda$ as a function of $n$. Theoretical guidelines for measuring this trade-off are left for future research, in our experiments, this is done on an empirical basis (we prove below a loose sufficient condition, where $\lambda^2 n^{1/2}$ has to be large).

Figure 3: Illustration of the overfitting phenomenon. The target density $p$ is represented in red. We approximate it by $d\rho_n$ supported on the black dots with corresponding weights $\eta_i$ proportional to $p(x_i)$, $i = 1, \ldots, n$. For $\lambda = 10^{-3}$, we use Eq. (2) to compute the corresponding empirical Christoffel function represented in dark blue.

**Monte Carlo approximation:**   Assuming that $\int_{\mathbb{R}^d} p(x)dx = 1$, if one has the possibility to draw an *i.i.d.* sample $(x_i)_{i=1,\ldots,n}$, with density $p$, then one can use $\eta_i = \frac{1}{n}$ for $i = 1, \ldots, n$. Our estimators take the form $\hat{C}_\lambda(z)^{-1} = \left\langle k(z,\cdot), (\hat{\Sigma} + \lambda I)^{-1} k(z,\cdot) \right\rangle_{\mathcal{H}}$, where $\hat{\Sigma}$ is the empirical covariance

operator. Thus, we have:

$$
\begin{aligned}
\left|\hat{C}_\lambda(z)^{-1} - C_\lambda(z)^{-1}\right| &\leq \left|\left\langle k(z,\cdot), \left[(\hat{\Sigma}+\lambda I)^{-1} - (\Sigma+\lambda I)^{-1}\right]k(z,\cdot)\right\rangle_{\mathcal{H}}\right| \\
&\leq \|k(z,\cdot)\|_{\mathcal{H}}^2 \|(\hat{\Sigma}+\lambda I)^{-1} - (\Sigma+\lambda I)^{-1}\|_{\mathrm{op}} \\
&\leq k(z,z)\lambda^{-2}\|\Sigma - \hat{\Sigma}\|_{\mathrm{op}}.
\end{aligned}
\tag{8}
$$

Since, $\|\Sigma - \hat{\Sigma}\|_{\mathrm{op}}$ is of order $n^{-1/2}$ (see, e.g., [25]), if $\lambda^2 n^{1/2}$ is large enough, then we obtain a good estimation of the Christoffel function (note that better bounds could be obtained with respect to $\lambda$ using tools from [6, 2, 28]).

**Gaussian kernel:** A natural question is whether Theorem 1 holds for the Gaussian kernel $k\colon (x,y) \mapsto e^{\frac{\|x-y\|^2}{l}}$ where $l > 0$ is a bandwidth parameter. For this choice of kernel, $D(\lambda)$ is of the order of $-1/\log(\lambda)$, which decreases very slowly. We conjecture that Assumption 2 fails in this setting and that Theorem 1 does not hold. Indeed, performing the same simulation as in Section 4 with a piecewise constant density, we observe that the localization phenomenon no longer holds. This is presented in Figure 4 which displays important boundary effects around discontinuities. For comparison purpose, Figure 5 gives the same result for Mattérn kernels.

Figure 4: Estimate of the Christoffel function for the Gaussian kernel, different values of the bandwidth and a piecewise constant density. The setting is the same as in the experiment presented in Section 4. The behavior at the discontinuities suggest that variations of the density affect the value of the Christoffel function *beyond* the local scale described in Theorem 1.

Figure 5: Estimate of the Christoffel function for the Mattèrn kernel, different values of the parameters and a piecewise constant density. The setting is the same as in the experiment presented in Section 4. The behavior at the discontinuities suggest that variations of the density affect the value of the Christoffel *only at* the local scale described in Theorem 1.

**Illustration in dimension 2:** For illustration purpose, we consider a density on the unit square in dimension 2 and compare it with the estimate obtained from the regularized Christoffel function

using the Riemann plug-in approximation procedure. We choose the Matérn kernel with $\nu = 1$ and $l = 0.2$ and $\lambda = 0.001$. Figure 6 illustrates the correspondence between the true density and the obtained estimate.

Figure 6: Comparison between the level sets of a given density (right) and the estimate given by the empirical Christoffel function with $\nu = 1$, $l = 0.2$ and $\lambda = 0.001$ (left). We use the Riemann plug-in approximation procedure with a grid of 2025 points on $[-1.5, 1.5]^2$. The estimate captures both the round shape of the level sets in the middle and the squared shape of the support ($[-1, 1]^2$).

## B    Proofs

### B.1    Proof of Theorem 1

The proof is organized as follows, first we will prove an upper bound on $C_\lambda$ which is of the order of the claimed equivalent plus negligible terms. In a second step we produce a lower bound on $C_\lambda$ which is of the same nature. Assumptions 1 and 2 are assumed to hold true throughout this section.

Recall that we have $f_\lambda = D(\lambda)g_\lambda$ with the notations given in Eq. (3) of the main text. We will work with $z = 0$ since the general result can be obtained by a simple translation. We consider $p$ as in Assumption 1 and assume throughout the section that $p(0) = 1$. This is without loss of generality since $p(0) > 0$, one can substitute $p$ by $p/p(0)$ and $\lambda$ by $\lambda/p(0)$ and use

$$C_\lambda(0) = \min_{f \in \mathcal{H}_0} \int_{\mathbb{R}^d} f(x)^2 p(x) dx + \lambda \|f\|_{\mathcal{H}}^2 = p(0) \min_{f \in \mathcal{H}_0} \int_{\mathbb{R}^d} f(x)^2 \frac{p(x)}{p(0)} dx + \frac{\lambda}{p(0)} \|f\|_{\mathcal{H}}^2. \quad (9)$$

Combining translations and scaling in (9), we only need to show that $C_\lambda(0) \sim D(\lambda)$ when $p(0) = 1$ and $p$ is continuous at 0.

**Upper bound:**    For any $\lambda > 0$, $f_\lambda$ is feasible for $C_\lambda(0)$ in problem (1) and therefore, using $p(0) = 1$,

$$C_\lambda(0) \leq \left( \int f_\lambda(x)^2 dx + \lambda \|f_\lambda\|_{\mathcal{H}}^2 \right) + \int_{\mathbb{R}^d} (p(x) - 1) f_\lambda^2(x) dx$$

$$= D(\lambda) + \int_{\mathbb{R}^d} (p(x) - 1) f_\lambda^2(x) dx. \quad (10)$$

We only need to control the last term. The result then follows from the next Lemma which proof is postponed to Section B.2.

**Lemma 3** *As $\lambda \to 0$ with $\lambda > 0$,*

$$\int_{\mathbb{R}^d} f_\lambda(x)^2 \left( p(x) - 1 \right) dx = o(D(\lambda)).$$

Combining (10) and Lemma 3, we obtain, as $\lambda \to 0$

$$C_\lambda(0) \leq D(\lambda) + o(D(\lambda)). \quad (11)$$

**Lower bound:** To prove the lower bound, let $E \colon t \mapsto \sup_{\|x\| \leq t} |p(x) - 1|$. The quantity $E$ is non negative and we have $\lim_{t \to 0} E(t) = 0$ by continuity of $p$. Choosing $\varepsilon(\lambda)$ as given by Assumption 2, we obtain for any $\lambda > 0$ sufficiently small, using $p(0) = 1$,

$$
\begin{aligned}
C_\lambda(0) &\geq \inf_{f \in \mathcal{H}_0} \int_{\|x\| \leq \varepsilon(\lambda)} f(x)^2 p(x) dx + \lambda \|f\|_{\mathcal{H}}^2 \\
&\geq \inf_{f \in \mathcal{H}_0} \left(1 - E\left(\varepsilon(\lambda)\right)\right) \int_{\|x\| \leq \varepsilon(\lambda)} f(x)^2 dx + \lambda \|f\|_{\mathcal{H}}^2 \\
&\geq \inf_{f \in \mathcal{H}_0} \left(1 - E\left(\varepsilon(\lambda)\right)\right) \left( \int_{\|x\| \leq \varepsilon(\lambda)} f(x)^2 dx + \lambda \|f\|_{\mathcal{H}}^2 \right).
\end{aligned}
\tag{12}
$$

We need to control the last term. This is the purpose of the following Lemma which proof is postponed to Section B.2.

**Lemma 4** *Let $\varepsilon$ be given as in Assumption 2, then, as $\lambda \to 0$ with $\lambda > 0$, we have*

$$
\inf_{f \in \mathcal{H}_0} \int_{\|x\| \leq \varepsilon(\lambda)} f(x)^2 dx + \lambda \|f\|_{\mathcal{H}}^2 \geq D(\lambda) + o(D(\lambda)).
$$

Combining (12) and Lemma 4, we obtain, as $\lambda \to 0$

$$
C_\lambda(0) \geq \left(1 - E\left(\varepsilon(\lambda)\right)\right) \left(D(\lambda) + o(D(\lambda))\right) = D(\lambda) + o(D(\lambda)).
\tag{13}
$$

To conclude, combining (11) and (13), we obtain $C_\lambda(0) \sim D(\lambda)$ as claimed.

## B.2 Lemmas for Section B.1 and proof of Lemma 1 of the main text

**Proof of Lemma 1:** Eq. (3) characterizes $\mathcal{H}$ and in particular, any function in $\mathcal{H}$ is in $L^2$ so that Parseval theorem holds. Furthermore for any $f \in \mathcal{H}$, $\hat{f}$ is in $L^2(\mathbb{R}^d) \cap L^1(\mathbb{R}^d)$ (see Remark 2). Rewriting (5) in the Fourier domain, we have

$$
\begin{aligned}
D(\lambda) = \quad &\inf \quad \frac{1}{(2\pi)^d} \int_{\mathbb{R}^d} |\hat{f}(\omega)|^2 \frac{\hat{q}(\omega) + \lambda}{\hat{q}(\omega)} d\omega \\
&\text{s.t.} \quad \hat{f} \in L^2(\mathbb{R}^d) \cap L^1(\mathbb{R}^d) \\
&\qquad \int_{\mathbb{R}^d} \frac{|\hat{f}(\omega)|^2}{\hat{q}(\omega)} d\omega < +\infty \\
&\qquad \frac{1}{(2\pi)^d} \int_{\mathbb{R}^d} \hat{f}(\omega) d\omega = 1.
\end{aligned}
\tag{14}
$$

The space $\tilde{\mathcal{H}} = \left\{ \hat{f} \in L^2(\mathbb{R}^d) \cap L^1(\mathbb{R}^d); \ \int_{\mathbb{R}^d} \frac{|\hat{f}(\omega)|^2}{\hat{q}(\omega)} d\omega < +\infty \right\}$ endowed with the inner product $\frac{1}{(2\pi)^d} \left\langle \hat{f}_1, \hat{f}_2 \right\rangle_{\tilde{\mathcal{H}}} = \int_{\mathbb{R}^d} \frac{\hat{f}_1(\omega) \overline{\hat{f}_2(\omega)}}{\hat{q}(\omega)} d\omega$ is a Hilbert space which is simply the image of $\mathcal{H}$ by the Fourier transform. Problem (14) can be rewritten in a form that fits Lemma 5 below as follows

$$
\begin{aligned}
D(\lambda) = \quad &\inf \quad \left\langle \hat{f}, M\hat{f} \right\rangle_{\tilde{\mathcal{H}}} \\
&\text{s.t.} \quad \hat{f} \in \tilde{\mathcal{H}} \\
&\qquad \left\langle \hat{f}, \hat{q} \right\rangle_{\tilde{\mathcal{H}}} = 1,
\end{aligned}
\tag{15}
$$

where $M$ is the operator which consists in multiplication by $(\hat{q} + \lambda)$. For any $\hat{f} \in \tilde{\mathcal{H}}$, we have $\|M\hat{f}\|_{\tilde{\mathcal{H}}}^2 \leq (\lambda + \|\hat{q}\|_{L^\infty(\mathbb{R}^d)})^2 \|\hat{f}\|_{\tilde{\mathcal{H}}}^2$ and $M$ is bounded on $\tilde{\mathcal{H}}$. Using Lemma 5, we get an expression for the solution of the minimization problem in (14) of the form

$$
\hat{f}(\omega) = D(\lambda) \frac{\hat{q}(\omega)}{\hat{q}(\omega) + \lambda},
$$

for all $\omega \in \mathbb{R}^d$, where the optimal value $D(\lambda)$, ensures that $\langle \hat{f}, \hat{q} \rangle_{\tilde{\mathcal{H}}} = 1$. We deduce the value of $D(\lambda)$ and get back to $\mathcal{H}$ by combining Eq. (3) with the inverse Fourier transform of $\hat{f}$ which leads to the claimed expression for $f_\lambda$. ∎

**Proof of Lemma 3:** Let $E\colon t \mapsto \sup_{\|x\| \leq t} |p(x) - 1|$. We have $\lim_{t \to 0} E(t) = 0$ by continuity of $p$. Let $\varepsilon(\lambda)$ be given as in Assumption 2.

$$\int_{\mathbb{R}^d} f_\lambda^2(x)\,(p(x) - 1)\,dx$$

$$= \int_{\|x\| \geq \varepsilon(\lambda)} f_\lambda^2(x)\,(p(x) - 1)\,dx + \int_{\|x\| \leq \varepsilon(\lambda)} f_\lambda^2(x)\,(p(x) - 1)\,dx$$

$$\leq \|p\|_{L^\infty(\mathbb{R}^d)} \int_{\|x\| \geq \varepsilon(\lambda)} f_\lambda^2(x)\,dx + E(\varepsilon(\lambda)) \int_{\|x\| \leq \varepsilon(\lambda)} f_\lambda^2(x)\,dx$$

$$\leq \|p\|_{L^\infty(\mathbb{R}^d)} \int_{\|x\| \geq \varepsilon(\lambda)} f_\lambda^2(x)\,dx + E(\varepsilon(\lambda)) D(\lambda).$$

Using Assumption 2, as $\lambda > 0$, the first term is $o(D(\lambda))$ and the sum is also $o(D(\lambda))$. This proves the desired result. ∎

**Proof of Lemma 4:** Consider the surrogate problem, for any $\lambda, \epsilon > 0$,

$$\tilde{D}_\epsilon(\lambda) = \inf_{g \in \mathcal{H}_0} \int_{\|x\| \leq \epsilon} g(x)^2 dx + \lambda \|g\|_{\mathcal{H}}^2.$$

From Eq. (4), we have for all $g \in \mathcal{H}$,

$$\|g\|_{\mathcal{H}}^2 = \frac{1}{(2\pi)^d} \int_{\mathbb{R}^d} \frac{|\hat{g}(\omega)|^2}{\hat{q}(\omega)} d\omega \geq \frac{1}{(2\pi)^d \|\hat{q}\|_\infty} \int_{\mathbb{R}^d} |\hat{g}(\omega)|^2 d\omega = \frac{1}{\|\hat{q}\|_\infty} \int_{\mathbb{R}^d} g(x)^2 dx, \qquad (16)$$

where we have used Parseval identity. Note that $\|\hat{q}\|_\infty$ is finite since $q$ is in $L_1$. We fix arbitrary $\lambda > 0$, $\epsilon > 0$ and denote by $B_\epsilon$ the Euclidean ball of radius $\epsilon$. For any $f, g \in \mathcal{H}$, we have using Cauchy-Schwartz inequality,

$$\left| \int_{\mathbb{R}^d} f(x)g(x)dx \right|^2 \leq \int_{\mathbb{R}^d} (f(x))^2 dx \int_{\mathbb{R}^d} (g(x))^2 dx \leq \|\hat{q}\|_{L^\infty(\mathbb{R}^d)}^2 \|f\|_{\mathcal{H}}^2 \|g\|_{\mathcal{H}}^2$$

$$\left| \int_{B_\epsilon} f(x)g(x)dx \right|^2 \leq \int_{B_\epsilon} (f(x))^2 dx \int_{B_\epsilon} (g(x))^2 dx \leq \|\hat{q}\|_{L^\infty(\mathbb{R}^d)}^2 \|f\|_{\mathcal{H}}^2 \|g\|_{\mathcal{H}}^2.$$

Hence both expressions define bounded symmetric bi-linear forms on $\mathcal{H}$ and there is a semidefinite bounded self adjoint operator associated to each of these forms. We call the corresponding operators $\Sigma\colon \mathcal{H} \mapsto \mathcal{H}$ and $M_\epsilon\colon \mathcal{H} \mapsto \mathcal{H}$ respectively, they satisfy for any $f, g \in \mathcal{H}$,

$$\langle f, \Sigma g \rangle_{\mathcal{H}} = \int_{\mathbb{R}^d} f(x)g(x)dx$$

$$\langle f, M_\epsilon g \rangle_{\mathcal{H}} = \int_{B_\epsilon} f(x)g(x)dx.$$

We can apply Lemma 5 and the solution for $\tilde{D}_\epsilon(\lambda)$ is proportional to $\tilde{g}_\lambda = (M_\epsilon + \lambda I)^{-1} K_0$ and the value of this problem is $\tilde{g}_\lambda(0)^{-1} = \frac{1}{\langle K_0, \tilde{g}_\lambda \rangle_{\mathcal{H}}}$ where $K_0 = k(0, \cdot) \in \mathcal{H}$. Similar reasoning hold for $g_\lambda$ and $D(\lambda)$. We have

$$D(\lambda)^{-1} - \tilde{D}_\epsilon(\lambda)^{-1} = g_\lambda(0) - \tilde{g}_\lambda(0) = \left\langle K_0, ((\Sigma + \lambda I)^{-1} - (M_\epsilon + \lambda I)^{-1}) K_0 \right\rangle_{\mathcal{H}}$$

$$= \left\langle K_0, (\Sigma + \lambda I)^{-1}(M_\epsilon - \Sigma)(M_\epsilon + \lambda I)^{-1} K_0 \right\rangle_{\mathcal{H}}$$

$$= \langle g_\lambda, M_\epsilon \tilde{g}_\lambda \rangle_{\mathcal{H}} - \langle g_\lambda, \Sigma \tilde{g}_\lambda \rangle_{\mathcal{H}}$$

$$= \int_{\|x\| \geq \epsilon} g_\lambda(x) \tilde{g}_\lambda(x) dx.$$

Hence, we obtain by Cauchy-Schwartz inequality

$$|g_\lambda(0) - \tilde{g}_\lambda(0)|^2 = \left( \int_{\|x\| \geq \epsilon} g_\lambda(x) \tilde{g}_\lambda(x) dx \right)^2$$

$$\leq \int_{\|x\| \geq \epsilon} g_\lambda(x)^2 dx \int_{\mathbb{R}^d} \tilde{g}_\lambda(x)^2 dx.$$

From (16), we deduce that

$$\int_{\mathbb{R}^d} \tilde{g}_\lambda^2(x) dx \leq \|\hat{q}\|_\infty \|\tilde{g}_\lambda\|_{\mathcal{H}}^2 = \|\hat{q}\|_\infty \tilde{D}_\epsilon(\lambda)^{-2} \|\tilde{f}_\lambda\|_{\mathcal{H}}^2 \leq \|\hat{q}\|_\infty \frac{1}{\lambda \tilde{D}_\epsilon(\lambda)},$$

and obtain,

$$|g_\lambda(0) - \tilde{g}_\lambda(0)|^2 \leq \|\hat{q}\|_\infty \tilde{D}_\epsilon(\lambda)^{-1} \lambda^{-1} \int_{\|x\| \geq \epsilon} g_\lambda(x)^2 dx$$

$$\leq \|\hat{q}\|_\infty \tilde{D}_\epsilon(\lambda)^{-1} \lambda^{-1} D(\lambda)^{-2} \int_{\|x\| \geq \epsilon} f_\lambda(x)^2 dx.$$

Now using Assumption 2, we can set $\epsilon = \varepsilon(\lambda)$, so that as $\lambda \to 0$, $|g_\lambda(0) - \tilde{g}_\lambda(0)|^2 = o(\tilde{D}_{\varepsilon(\lambda)}(\lambda)^{-1} D(\lambda)^{-1})$ and, using $\tilde{D}_{\varepsilon(\lambda)} \leq D(\lambda)$,

$$|D(\lambda) - \tilde{D}_{\varepsilon(\lambda)}(\lambda)| = o\left( \sqrt{\tilde{D}_{\varepsilon(\lambda)}(\lambda) D(\lambda)} \right) = o(D(\lambda)).$$

Hence, as $\lambda \to 0$, we obtain $\tilde{D}_{\varepsilon(\lambda)}(\lambda) \geq D(\lambda) + o(D(\lambda))$ as claimed. ∎

## B.3 Proof of Theorem 2

Similarly as in Section B.1, we assume that $z = 0$, and there exists $\epsilon > 0$ such that $\int_{\|x\| \leq \epsilon} p(x) dx = 0$. In this case, we have for any $\lambda'$ such that $\varepsilon(\lambda') \leq \epsilon$ and any $\lambda > 0$,

$$C_\lambda(0) = \inf_{f \in \mathcal{H}_0} \int_{\mathbb{R}^d} f(x)^2 p(x) dx + \lambda \|f\|_{\mathcal{H}}^2$$

$$\leq \int_{\mathbb{R}^d} p(x) f_{\lambda'}(x)^2 dx + \lambda \|f_{\lambda'}\|_{\mathcal{H}}^2$$

$$\leq \|p\|_\infty \int_{\|x\| \geq \varepsilon(\lambda')} f_{\lambda'}(x)^2 dx + \lambda \|f_{\lambda'}\|_{\mathcal{H}}^2$$

$$\leq o(D(\lambda')\lambda') + (\lambda/\lambda') D(\lambda').$$

Taking $\lambda' = \sqrt{\lambda}$ proves case (i).

Case (ii) in Theorem 2 follows from a simple argument, using the variational formulation in (1). Consider a $C^\infty$ function which evaluates to 1 at 0 and to 0 outside of the ball of radius $\epsilon$ centered at 0. Call this function $f_\epsilon$. This function is feasible for problem (1) for any value of $\lambda$ and hence we have $C_\lambda(z) \leq \lambda \|f_\epsilon\|_{\mathcal{H}}$, for all $\lambda > 0$. Note that it follows from Eq. (3) that $\|f_\epsilon\|_{\mathcal{H}}$ must be finite since $f_\epsilon$ is $C^\infty$ which implies that $\hat{f}_\epsilon$ is decreasing to 0 faster than any polynomial at infinity and our added assumptions on the kernel imply that $\hat{f}_\epsilon$ is in $\mathcal{H}$.

## B.4 Proofs for Section 3.3

**Proof of Lemma 2 (i):** Lemma 1 provides an analytic description of $D(\lambda)$ and a characterization of the solution $f_\lambda$. We prove the asymptotic expansion of $D(\lambda)$ as $\lambda \to 0$. We have

$$\hat{g}_\lambda \colon \omega \mapsto \frac{1}{1 + \lambda (P(\omega))^\gamma},$$

and hence, denoting by $R$, the polynomial $P - Q$ which is of degree at most $2s - 1$, for any $x \in \mathbb{R}^d$,

$$
\begin{aligned}
g_\lambda(x) &= \frac{1}{(2\pi)^d} \int_{\mathbb{R}^d} \frac{e^{ix^\top \omega}}{1 + \lambda(P(\omega))^\gamma} d\omega \\
&= \frac{\lambda^{-d/(2s\gamma)}}{(2\pi)^d} \int_{\mathbb{R}^d} \frac{e^{i(x\lambda^{-1/(2s\gamma)})^\top(\omega\lambda^{1/(2s\gamma)})}}{1 + \left(\lambda^{\frac{1}{\gamma}} R(\omega) + Q(\omega\lambda^{1/(2s\gamma)})\right)^\gamma} \lambda^{d/(2s\gamma)} d\omega \\
&= \frac{\lambda^{-d/(2s\gamma)}}{(2\pi)^d} \int_{\mathbb{R}^d} \frac{e^{i(x\lambda^{-1/(2s\gamma)})^\top \omega}}{1 + \left(\lambda^{\frac{1}{\gamma}} R(\omega\lambda^{-1/(2s\gamma)}) + Q(\omega)\right)^\gamma} d\omega.
\end{aligned}
\tag{17}
$$

We deduce the following

$$
\begin{aligned}
\left|g_\lambda(0) - \lambda^{-d/(2s\gamma)} q_0\right| &= \left| \frac{\lambda^{-d/(2s\gamma)}}{(2\pi)^d} \int_{\mathbb{R}^d} \left( \frac{1}{1 + \left(\lambda^{\frac{1}{\gamma}} R(\omega\lambda^{-1/(2s\gamma)}) + Q(\omega)\right)^\gamma} - \frac{1}{1 + (Q(\omega))^\gamma} \right) d\omega \right| \\
&= \frac{\lambda^{-d/(2s\gamma)}}{(2\pi)^d} \int_{\mathbb{R}^d} \frac{\left(\lambda^{\frac{1}{\gamma}} R(\omega\lambda^{-1/(2s\gamma)}) + Q(\omega)\right)^\gamma - (Q(\omega))^\gamma}{\left(1 + \left(\lambda^{\frac{1}{\gamma}} R(\omega\lambda^{-1/(2s\gamma)}) + Q(\omega)\right)^\gamma\right)(1 + (Q(\omega))^\gamma)} d\omega \\
&\leq \frac{\lambda^{-d/(2s\gamma)}}{(2\pi)^d} \int_{\mathbb{R}^d} \frac{\gamma \left(\lambda^{\frac{1}{\gamma}} R(\omega\lambda^{-1/(2s\gamma)}) + Q(\omega)\right)^{\gamma-1} \lambda^{\frac{1}{\gamma}} R(\omega\lambda^{-1/(2s\gamma)})}{\left(1 + \left(\lambda^{\frac{1}{\gamma}} R(\omega\lambda^{-1/(2s\gamma)}) + Q(\omega)\right)^\gamma\right)(1 + (Q(\omega))^\gamma)} d\omega,
\end{aligned}
\tag{18}
$$

where we have used the fact that for any $x, y \geq 0$, $(x+y)^\gamma \leq \gamma(x+y)^{\gamma-1}x + y^\gamma$ which is a direct application of Taylor-Lagrange inequality. Now consider a constant, $M$, as given by Lemma 6 such that

$$
R \leq M(1 + Q^{\frac{2s-1}{2s}}).
$$

We have for any $\lambda > 0$ and any $\omega \in \mathbb{R}^d$,

$$
\begin{aligned}
\lambda^{\frac{1}{\gamma}} R(\omega\lambda^{-1/(2s\gamma)}) &\leq M \left(\lambda^{\frac{1}{\gamma}} + \lambda^{\frac{1}{\gamma}} Q(\omega\lambda^{-1/(2s\gamma)})^{\frac{2s-1}{2s}}\right) \\
&= M \left(\lambda^{\frac{1}{\gamma}} + \lambda^{\frac{1}{2s\gamma}} Q(\omega)^{\frac{2s-1}{2s}}\right) \\
&= M\lambda^{\frac{1}{2s\gamma}} \left(\lambda^{\frac{2s-1}{2s\gamma}} + Q(\omega)^{\frac{2s-1}{2s}}\right) \\
&\leq M 2^{\frac{1}{2s}} \lambda^{\frac{1}{2s\gamma}} \left(\lambda^{\frac{1}{\gamma}} + Q(\omega)\right)^{\frac{2s-1}{2s}} \\
&\leq M 2^{\frac{1}{2s}} \lambda^{\frac{1}{2s\gamma}} \left(\lambda^{\frac{1}{\gamma}} R(\omega\lambda^{-1/(2s\gamma)}) + Q(\omega)\right)^{\frac{2s-1}{2s}},
\end{aligned}
\tag{19}
$$

where we have used Jensen's inequality and the fact that $R \geq 1$ from Assumption 3 for the last two identities. Combining (18) and (19), we obtain for any $\lambda \geq 0$

$$
\left|g_\lambda(0) - \lambda^{-d/(2s\gamma)} q_0\right| \leq \lambda^{\frac{1-d}{2s\gamma}} \frac{M\gamma 2^{\frac{1}{2s}}}{(2\pi)^d} \int_{\mathbb{R}^d} \frac{\left(\lambda^{\frac{1}{\gamma}} R(\omega\lambda^{-1/(2s\gamma)}) + Q(\omega)\right)^{\gamma-\frac{1}{2s}}}{\left(1 + \left(\lambda^{\frac{1}{\gamma}} R(\omega\lambda^{-1/(2s\gamma)}) + Q(\omega)\right)^\gamma\right)(1 + (Q(\omega))^\gamma)} d\omega.
\tag{20}
$$

A standard computation ensures that for any $x \geq 0$

$$
\frac{x^{\gamma-\frac{1}{2s}}}{1 + x^\gamma} = \frac{(x^\gamma)^{1-\frac{1}{2s\gamma}}}{1 + x^\gamma} \leq \frac{(1+x^\gamma)^{1-\frac{1}{2s\gamma}}}{1 + x^\gamma} = (1+x^\gamma)^{-\frac{1}{2s\gamma}} \leq 1.
\tag{21}
$$

Combining (20) and (21) we obtain for all $\lambda > 0$,

$$
\left|g_\lambda(0) - \lambda^{-d/(2s\gamma)} q_0\right| \leq \lambda^{\frac{1-d}{2s\gamma}} \frac{M\gamma 2^{\frac{1}{2s}}}{(2\pi)^d} \int_{\mathbb{R}^d} \frac{1}{(1 + (Q(\omega))^\gamma)} d\omega.
\tag{22}
$$

In particular, we deduce from (22) that

$$\frac{1}{D(\lambda)} = g_\lambda(0) = q_0 \lambda^{-d/(2s\gamma)} + O(\lambda^{(1-d)/(2s\gamma)}).$$

So that $D(\lambda) = \frac{\lambda^{d/(2s\gamma)}}{q_0} + O(\lambda^{(1+d)/(2s\gamma)})$ which is the desired result.

$\blacksquare$

**Proof of Lemma 2 (ii):** We verify that the choice of $p \in \mathbb{N}^*$ ensures that $2p \in [2s\gamma, 4s\gamma)$. Indeed, if $s\gamma \geq 1$, we have by definition of the upper integral part that $2s\gamma \leq 2p < 2s\gamma + 2 \leq 4s\gamma$. If $s\gamma < 1$, we have $p = 1$ and $2s\gamma > d \geq 1$ so that $2s\gamma \leq 2p = 2 < 4s\gamma$. We deduce from Lemma 9 that there exists a constant $N$ such that for any $\omega \in \mathbb{R}^d$ and $\lambda > 0$,

$$\left| \frac{\partial^{2p} \hat{g}_\lambda(\omega)}{\partial \omega_1^{2p}} \right| \leq N\lambda \hat{g}_\lambda(\omega). \tag{23}$$

Hence successive derivatives of $\hat{g}_\lambda$ are in $L^1$. Differentiating under the integral sign for the Fourier transform ensures that differentiation in the Fourier domain amounts to multiplication by a monomial in the space domain, we obtain the following bound:

$$\sup_{x \in \mathbb{R}^d} |x_1^{2p} g_\lambda(x)| \leq \frac{1}{(2\pi)^d} \int_{\mathbb{R}^d} \left| \frac{\partial^{2p} \hat{g}_\lambda(\omega)}{\partial \omega_1^{2p}} \right| d\omega.$$

Evaluating the inverse Fourier transform of $\hat{g}_\lambda$ at 0, using (23), we have for any $\lambda > 0$

$$\sup_{x \in \mathbb{R}^d} |x_1^{2p} g_\lambda(x)| \leq \frac{N\lambda}{D(\lambda)},$$

and

$$\sup_{x \in \mathbb{R}^d} |x_1^{2p} f_\lambda(x)| \leq N\lambda.$$

The choice of $x_1$ was arbitrary and similar results hold for all coordinates. We deduce that there exists $M_1 > 0$ such that for all $x \in \mathbb{R}^d$ and $\lambda > 0$

$$f_\lambda^2(x) \frac{\|x\|^{8p}}{1 + \|x\|^{6p}} \leq M_1 \lambda^2 \frac{\|x\|^{4p}}{1 + \|x\|^{6p}}.$$

Note that the right hand side function is integrable since $2p \geq 2s\gamma > d$. We have for any $l > 0$ and $\lambda \in (0, 1)$,

$$\int_{\|x\| \geq \lambda^l} f_\lambda^2(x) \frac{\|x\|^{8p}}{1 + \|x\|^{6p}} dx \geq \frac{\lambda^{8pl}}{1 + \lambda^{6pl}} \int_{\|x\| \geq \lambda^l} f_\lambda^2(x) dx \geq \frac{\lambda^{8pl}}{2} \int_{\|x\| \geq \lambda^l} f_\lambda^2(x) dx$$

Combining the last two inequalities, we obtain

$$\int_{\|x\| \geq \lambda^l} f_\lambda^2(x) dx \leq M_2 \lambda^{2-8pl},$$

for some constant $M_2$ and any $\lambda \in (0, 1)$. Choosing $l < \left(1 - \frac{d}{2s\gamma}\right)/(8p)$ ensures that $2 - 8pl > 1 + \frac{d}{2s\gamma}$ and hence $\lambda^{2-8pl} = o(\lambda D(\lambda))$, using Lemma 2 (i). This is the desired result. $\blacksquare$

## C  Additional Lemmas

**Lemma 5** *Let $H$ be a complex Hilbert space with Hermitian form $\langle \cdot, \cdot \rangle_H$, let $M \colon H \to H$ be a bounded Hermitian invertible and positive operator and let $u \in H$. Then*

$$\frac{1}{\langle u, M^{-1}u \rangle_H} = \min_{x \in H} \quad \langle x, Mx \rangle_H$$

$$\text{s.t.} \quad \langle x, u \rangle_H = 1,$$

*and the optimal value is attained for $x_0 = \frac{M^{-1}u}{\langle u, M^{-1}u \rangle_H}$.*

**Proof** For any $y \in H$, we have

$$
\begin{aligned}
\langle y, My \rangle_H &= \langle y - x_0 + x_0, M(y - x_0 + x_0) \rangle_H \\
&= \langle y - x_0, M(y - x_0) \rangle_H + \langle x_0, Mx_0 \rangle_H + \langle y - x_0, Mx_0 \rangle_H + \langle Mx_0, y - x_0 \rangle_H \\
&= \langle y - x_0, M(y - x_0) \rangle_H + \frac{1}{\langle u, M^{-1}u \rangle_H} \left( 1 + \langle y - x_0, u \rangle_H + \langle u, y - x_0 \rangle_H \right) \\
&\geq \frac{1}{\langle u, M^{-1}u \rangle_H} \left( 1 + \langle y - x_0, u \rangle_H + \langle u, y - x_0 \rangle_H \right).
\end{aligned}
\tag{24}
$$

Now assume that $y$ is feasible, that is $\langle y, u \rangle_H = 1$, since $x_0$ is also feasible, we have $\langle y - x_0, u \rangle_H = \langle u, y - x_0 \rangle_H = 0$. This observation combined with the last inequality (24) concludes the proof. $\blacksquare$

**Lemma 6** *Let $P$ be a $2s$-positive $d$-variate polynomial as given in Definition 2 and let $Q$ be its $2s$-homogeneous part. Let $T$ be a $d$-variate polynomial of degree at most $t \in \mathbb{N}$. Then there exists a positive constant $M$ such that*

$$
T \leq M \left( 1 + Q^{\frac{t}{2s}} \right)
$$

$$
T \leq M P^{\frac{t}{2s}}.
$$

**Proof** Consider the following quantity

$$
M_1 = \max_{\omega \in \mathbb{R}^d, \, \omega \neq 0} \frac{\|\omega\|_\infty}{Q(\omega)^{\frac{1}{2s}}},
\tag{25}
$$

Note that, this quantity is well defined because the objective function is homogeneous of degree $0$, which means invariant by positive scaling. Furthermore, we have for all $\omega \in \mathbb{R}^d$, that $\|\omega\|_\infty \leq M_1 Q(\omega)^{\frac{1}{2s}}$. Now consider any monomial of the form $\omega^\beta$ for some multi index $\beta \in \mathbb{N}^d$, with $|\beta| \leq t$, we have for any $\omega$

$$
|\omega^\beta| \leq \|\omega\|_\infty^{|\beta|} \leq M_1^{|\beta|} Q(\omega)^{\frac{|\beta|}{2s}} \leq M_1^{|\beta|} \left( 1 + Q(\omega)^{\frac{t}{2s}} \right),
$$

Since $T$ is of degree at most $t$, this must hold for all the monomials of $T$. The first result follows by a simple summation over monomials of degree up to $t$. The second result follows similarly by using

$$
|\omega^\beta| \leq \|\omega\|_\infty^{|\beta|} \leq M_1^{|\beta|} Q(\omega)^{\frac{|\beta|}{2s}} \leq M_1^{|\beta|} P(\omega)^{\frac{|\beta|}{2s}} \leq M_1^{|\beta|} P(\omega)^{\frac{t}{2s}},
$$

where the last inequality holds because $P \geq 1$. $\blacksquare$

**Lemma 7 : Faà di Bruno Formula.** *Let $f \colon (0, +\infty) \mapsto \mathbb{R}$ and $g \colon \mathbb{R}^d \mapsto [1, +\infty)$ be infinitely differentiable functions. Then we have for any $n \in \mathbb{N}^*$*

$$
\frac{\partial^n}{\partial x_1^n} f \circ g(x) = \sum_{\pi \in \Pi} f^{(|\pi|)}(g(x)) \prod_{B \in \pi} \frac{\partial^{|B|} g}{\partial x_1^{|B|}}(x),
$$

*where $\Pi$ denotes all partitions of $\{1, \ldots, n\}$, the product is over subsets of $\{1, \ldots, n\}$ given by the partition $\pi$ and $|\cdot|$ denotes the number of elements of a set. We rewrite this as follows*

$$
\frac{\partial^n}{\partial x_1^n} f \circ g(x) = \sum_{k=1}^n \sum_{\pi \in \Pi_k} f^{(k)}(g(x)) \prod_{B \in \pi} \frac{\partial^{|B|} g}{\partial x_1^{|B|}}(x),
$$

*where $\Pi_k$ denotes all partitions of size $k$ of $\{1, \ldots, n\}$.*

**Proof** This is a special case of the result stated in [15, Propositions 1 and 2]. $\blacksquare$

**Lemma 8** *Let $P$ be a $2s$-positive $d$-variate polynomial as given in Definition 2 and let $\gamma \geq 1$ and $m \in \mathbb{N}^*$. Then there exists a positive constant $M_m$, such that*

$$\frac{\partial^m}{\partial x_1^m} P^\gamma \leq M_m P^{\gamma - \frac{m}{2s}}.$$

**Proof** We apply Lemma 7 with $f = (\cdot)^\gamma$ and $g = P$. We fix $k \in \{1, \ldots, m\}$, and $\pi$ a partition of $\{1, \ldots, m\}$ of size $k$. The $i$-th derivative of $P$ is a polynomial of degree at most $2s - i$. Hence the quantity

$$\prod_{B \in \pi} \frac{\partial^{|B|} P}{\partial x_1^{|B|}}$$

is a $d$-variate polynomial of degree at most $2sk - m$, because $\pi$ is of size $k$ and $\sum_{B \in \pi} |B| = m$ since $\pi$ is partition of $\{1, \ldots, m\}$. Using Lemma 6, there exists a constant $M_\pi$ such that

$$\prod_{B \in \pi} \frac{\partial^{|B|} P}{\partial x_1^{|B|}}(x) \leq M_\pi P^{k - \frac{m}{2s}}.$$

Using Lemma 7, we have

$$
\begin{aligned}
\frac{\partial^m}{\partial x_1^m} P^\gamma &= \sum_{k=1}^m \sum_{\pi \in \Pi_k} \left( \prod_{i=0}^{k-1} (\gamma - i) \right) P^{\gamma - k} \prod_{B \in \pi} \frac{\partial^{|B|} P}{\partial x_1^{|B|}} \\
&\leq \sum_{k=1}^m \sum_{\pi \in \Pi_k} \left| \prod_{i=0}^{k-1} (\gamma - i) \right| P^{\gamma - k} M_\pi P^{k - \frac{m}{2s}} \\
&= P^{\gamma - \frac{m}{2s}} \sum_{k=1}^m \sum_{\pi \in \Pi_k} \left| \prod_{i=0}^{k-1} (\gamma - i) \right| M_\pi,
\end{aligned}
$$

which is the desired result. ∎

**Lemma 9** *Let $P$ be a $2s$-positive $d$ variate polynomial as given in Definition 2 and let $\gamma \geq 1$. For any integer $n \in [2s\gamma, 4s\gamma)$, there exists a positive constant $N_n$, such that for any $\lambda > 0$,*

$$\frac{\partial^n}{\partial x_1^n} \left( \frac{1}{1 + \lambda P^\gamma} \right) \leq \frac{\lambda N_n}{1 + \lambda P^\gamma}.$$

**Proof** We fix $n \in \mathbb{N}^*$ and $\lambda > 0$. We apply Lemma 7 with $f \colon x \mapsto \frac{1}{1 + \lambda x}$ and $g = P^\gamma$, we obtain

$$\frac{\partial^n}{\partial x_1^n} \left( \frac{1}{1 + \lambda P^\gamma} \right) = \sum_{k=1}^n \sum_{\pi \in \Pi_k} \frac{1}{1 + \lambda P^\gamma} \frac{\lambda^k \prod_{i=1}^k (-i)}{(1 + \lambda P^\gamma)^k} \prod_{B \in \pi} \frac{\partial^{|B|} g}{\partial x_1^{|B|}}(x).$$

Applying Lemma 8 we obtain constants $M_1, \ldots, M_n$, such that,

$$
\begin{aligned}
\frac{\partial^n}{\partial x_1^n} \left( \frac{1}{1 + \lambda P^\gamma} \right) &\leq \sum_{k=1}^n \sum_{\pi \in \Pi_k} \frac{1}{1 + \lambda P^\gamma} \frac{\lambda^k k!}{(1 + \lambda P^\gamma)^k} \prod_{B \in \pi} M_{|B|} P^{\gamma - \frac{|B|}{2s}} \\
&= \sum_{k=1}^n \sum_{\pi \in \Pi_k} \frac{1}{1 + \lambda P^\gamma} \frac{\lambda^k k!}{(1 + \lambda P^\gamma)^k} P^{k\gamma - \frac{n}{2s}} \prod_{B \in \pi} M_{|B|} \\
&= \frac{\lambda^{\frac{n}{2s\gamma}}}{1 + \lambda P^\gamma} \sum_{k=1}^n \sum_{\pi \in \Pi_k} \frac{(\lambda P^\gamma)^{k - \frac{n}{2s\gamma}}}{(1 + \lambda P^\gamma)^k} k! \prod_{B \in \pi} M_{|B|},
\end{aligned}
$$

A standard computation gives for any $x \geq 0$ and $k \geq 2$, using the fact that $\frac{n}{2s\gamma} < 2$,

$$\frac{x^{k-\frac{n}{2s\gamma}}}{(1+x)^k} \leq \frac{(1+x)^{k-\frac{n}{2s\gamma}}}{(1+x)^k} = (1+x)^{-\frac{n}{2s\gamma}} \leq 1.$$

For $k = 1$, we have

$$\frac{(\lambda P^\gamma)^{1-\frac{n}{2s\gamma}}}{(1+\lambda P^\gamma)} \leq \lambda^{1-\frac{n}{2s\gamma}},$$

because $1 - \frac{n}{2s\gamma} \leq 0$ and $P \geq 1$. We deduce that

$$\frac{\partial^n}{\partial x_1^n}\left(\frac{1}{1+\lambda P^\gamma}\right) \leq \frac{\lambda M_n}{1+\lambda P^\gamma} + \frac{\lambda^{\frac{n}{2s\gamma}}}{1+\lambda P^\gamma}\sum_{k=2}^{n}\sum_{\pi \in \Pi_k} k! \prod_{B \in \pi} M_{|B|}.$$

This is the desired result since none of the constants depend on $\lambda$ and $n \geq 2s\gamma$ so that the leading term in the numerator is $O(\lambda)$. ∎