[Reviews · NeurIPS 2018]

Reviewer 1



The paper proposes estimating statistical leverage scores by regularized Christoffel scores. It develops Christoffel functions for kernel methods and tries to characterize the relationship between population density and leverage scores. The results are original and the paper is easy to read. However the significance is hard to gauge. It would have been nice if the authors could have taken the approach and applied it to Sketching or a related task where statistical leveraging is important. Such an application would have illustrated the significance of the obtained results.

Reviewer 2



*** UPDATE *** Thank you for your author feedback. I feel that the paper is strong, but discussion with fellow reviewers persuaded me to reduce my score from 9 to 8 due to their feedback that aspects of the presentation are unclear. *** This paper makes a strong mathematical contribution to the concept of leverage, linking this to notions from the theory of reproducing kernels and, ultimately, proposing and studying (mathematically) a "regularised" Christoffel function. Such a contribution is, to my mind, timely and its theoretical emphasis is welcome at NIPS. The paper reads well and the only comments that I have are extremely minor: - for transpose in latex, ^T ought to be ^\top - on p2 "which degree is" -> "whose degree is" - on p4 "approximate integration" -> "approximates integration" and "we first precise" -> "we first make precise" - on p4, technically L^\infty hasn't been defined. - the overview of numerical cubature in methods in sec 2.4 was rather limited, and the authors could emphasise that in general a cubature rule should be selected based on properties of the function being integrated. For example, arbitrarily fast convergence for smooth integrands can sometimes be achieved with quasi-Monte Carlo methods. - on p7 "numerical evidences" -> "numerical evidence"

Reviewer 3



The connection between leverage scores and density using the proposed regularized Christoffel functions is interesting. However, we found this work missing some motivations, namely the authors need to motivate the use of the proposed Christoffel functions compared to the commonly used methods in estimating leverage scores. We think that the constraint f(z)=1 in (1), and almost everywhere, needs to be further investigated, and its influence on the obtained results. For instance, the result given in (2) relies on the representer theorem, and the influence of this constraint does not show in this expression. This goes also for the other results. In the section 3.2 "Main result", the constraint in (5), namely f(0)=1, is not correct (the function f being defined on R^d).